# Principal Component Weighted Index for Wastewater Quality Monitoring

**Petr Praus**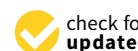

Institute of Environmental Technology, VŠB-Technical University of Ostrava, 17. listopadu 2172/15, 70800 Ostrava-Poruba, Czech Republic; petr.praus@vsb.cz

**Abstract:** The quality of raw and treated wastewater was evaluated using the principal component weighted index (PCWI) which was defined as a sum of principal component scores weighted according to their eigenvalues. For this purpose, five principal components (PCs) explaining 88% and 83% of the total variability of raw and treated wastewater samples, respectively, were extracted from 11 original physico-chemical parameters by robust principal component analysis (PCA). The PCWIs of raw and treated wastewater were analyzed in terms of their statistical distributions, temporal changes, mutual correlations, correlations with original parameters, and common water quality indexes (WQI). The PCWI allowed us to monitor temporal wastewater quality by one parameter instead of several. Unlike other weighted indexes, the PCWI is composed of independent variables with minimal information noise and objectively determined weights.

**Keywords:** principal component weighted index; wastewater quality; robust principal component analysis

## 1. Introduction

George E.P. Box has already expressed his skepticism of describing reality by mathematical models by his statement "All models are wrong, some are useful" [1]. In spite of this, various simple and easy-to-use composite indexes have been utilized to describe complex technical and scientific problems. For instance, there are social composite indexes [2,3], economic and well-being indexes [4,5], medical indexes [6,7], soil quality indexes [8,9], environmental quality [10–12], air quality indexes [13,14] and agriculture indexes [15], as well as water quality indexes [16–30] and so forth.

The composite indexes are based on a principle of the simple additive weighting (SAW) method combining independent criteria of which importance are expressed by their statistical weights [31]. These appropriate weights can be determined by subjective and objective methods. Subjective methods estimate the weights based on expert opinions and judgments of decision makers [32,33] or recommended standards [34]. A typical representation of this approach is the analytical hierarchy process (AHP) developed by Saaty [35]. Other methods are described in the literature [36,37]. The determination of objective weights is based on the application of various statistical measures, such as variably [38,39], correlation [40,41], and information content [42].

The above given requirements on the SAW model are in line with the basic properties of principal components created by principal component analysis (PCA): (i) the components are orthogonal and thus independent, and (ii) the components' weights correspond to their eigenvalues. Therefore, principal components were already used to construct various composite indexes characterizing the socioeconomic situation [43], soil quality [11], environment assessment [12], and surface water quality [44].

The aim of this paper is to demonstrate the utilization of principal component weighted index (PCWI) for the monitoring of raw and treated wastewater quality, which has never been described in the literature. Wastewater quality is of high importance nowadays because it is associated with the

release of a high quantity of contaminants such as dyes [45], pesticides and surfactants [46], heavy metals [47], etc. Such an index is assumed to be mathematically correct due to the use of independent variables (principal components) with minimal data noise and objective weights.

## 2. Materials and Methods

### 2.1. Data Collection

The wastewater samples were collected at an inlet and outlet of a mechanico-biological wastewater treatment plant (BWWTP) for the sake of regular monitoring of the treatment process. The BWWTP of type HYDROVIT 650-S (Vítkovice, Ostrava, Czech Republic) was designed for the treatment of common municipal wastewaters of 2100 population equivalents. The working area of the BWWTP was made up of a single overhead glass-fused to steel tank with a discontinuous cleaning process. A designed cleaning efficiency relating to biochemical oxygen demand (BOD) of 10–15 mg/L was 93%–95%, and an average flow of incoming waters was 650 m$^3$/day. The incoming and outgoing water was collected manually once a month (during one hour on the same day). The basic statistics of the samples' compositions are summarized in Tables 1 and 2.

**Table 1.** Summary statistics of raw wastewaters (n = 67).

| | $NH_4^+$ mg/L | BOD mg/L | COD mg/L | $NO_3^-$ mg/L | $NO_2^-$ mg/L | $PO_4^{3-}$ mg/L | TN mg/L | TSS mg/L | TP mg/L | pH mg/L | TDS mg/L |
|---|---|---|---|---|---|---|---|---|---|---|---|
| Aver. | 45.5 | 96.2 | 205 | 3.40 | 1.19 | 19.9 | 42.9 | 92 | 7.86 | 7.64 | 569 |
| St. dev. | 15.0 | 39.1 | 64.9 | 6.43 | 1.92 | 6.70 | 11.0 | 35 | 2.38 | 0.23 | 200 |
| Min. | 12.6 | 18.0 | 80.8 | 0.25 | 0.034 | 4.00 | 20.2 | 35 | 2.24 | 7.07 | 231 |
| Max. | 81.7 | 209 | 375 | 37.6 | 8.44 | 30.9 | 65.8 | 187 | 13.0 | 8.10 | 1532 |
| Median | 42.0 | 99.2 | 207 | 0.96 | 0.225 | 19.9 | 41.2 | 92 | 7.71 | 7.66 | 527 |
| Skew. | 0.435 | 0.063 | 0.212 | 3.34 | 2.12 | −0.144 | 0.255 | 0.510 | −0.171 | −0.301 | 2.48 |
| Kurt. | −0.217 | −0.185 | −0.477 | 12.2 | 3.71 | −0.638 | −0.477 | −0.091 | −0.228 | −0.312 | 8.15 |

**Table 2.** Summary statistics of treated wastewaters (n = 67).

| | $NH_4^+$ mg/L | BOD mg/L | COD mg/L | $NO_3^-$ mg/L | $NO_2^-$ mg/L | $PO_4^{3-}$ mg/L | TN mg/L | TSS mg/L | TP mg/L | pH mg/L | TDS mg/L |
|---|---|---|---|---|---|---|---|---|---|---|---|
| Aver. | 5.24 | 3.1 | 28.1 | 74.8 | 0.595 | 17.3 | 24.8 | 5 | 6.12 | 7.18 | 587 |
| St. dev. | 8.58 | 2.3 | 18.4 | 37.0 | 1.77 | 6.97 | 6.75 | 4 | 2.35 | 0.40 | 197 |
| Min. | 0.031 | 1.0 | 10.7 | 0.67 | 0.012 | 2.67 | 11.0 | 0 | 0.94 | 6.21 | 302 |
| Max. | 48.4 | 12.5 | 161 | 159 | 14.0 | 30.7 | 42.0 | 19 | 10.9 | 8.00 | 1580 |
| Median | 1.37 | 2.6 | 25.1 | 75.4 | 0.242 | 16.80 | 24.3 | 4 | 6.10 | 7.24 | 551 |
| Skew. | 2.76 | 2.40 | 5.74 | −0.045 | 6.78 | −0.071 | 0.567 | 1.47 | −0.155 | −0.334 | 2.30 |
| Kurt. | 9.17 | 6.43 | 38.9 | −0.373 | 48.0 | −0.803 | −0.105 | 1.68 | −0.580 | −0.581 | 8.36 |

The 67 raw and treated wastewater samples were characterized by 11 physico-chemical parameters, such as BOD after 5 days, chemical oxygen demand by dichromate (COD), total phosphorus (TP), total nitrogen (TN), total suspended solids (TSS), total dissolved salts (TDS), pH, ammonium, nitrate, nitrite, and phosphate. Water analyses including sampling and preservation were performed according to ISO and EN standard procedures: EN 1899-1: 1998 (BOD), ISO 6060: 1989 (COD), EN ISO 6878: 2004 (TP and phosphate), EN 25663:1993 (TN), EN 872:1996 (TSS and TDS), ISO 10523:2008 (pH), ISO 7150-1:1984 (ammonium), ISO 7890-3:1988 (nitrate), and ISO 6777:1984 (nitrite). The spectrophotometric determination of ammonium, nitrate, nitrite and phosphate were performed using a UV–VIS spectrometer DR 4200 (HACH Company, Loveland, CO, USA). TDS and TSS were determined gravimetrically after samples had been filtered through 0.85 μm membrane filters. pH was determined using a device pH 197 (WTW, Weilheim, Germany).

## 2.2. Robust Principal Component Analysis

Principal component analysis looks for new latent variables of $n$ samples, which are orthogonal (not correlated) to each other [48]. Each latent variable principal component is a linear combination of $p$ variables $x_i$ and describes a different source of total variation

$$X = TWT + E = \text{Data structure (model)} + \text{Noise} \qquad (1)$$

where X (n × m) is the data matrix, T (n × p) and W (m × p) are the matrixes of principal components scores and loadings, respectively, and E (n × m) is the residual matrix. Classical PCA can be performed by the eigenvalue decomposition of a correlation matrix. Robust PCA was performed by the eigenvalue decomposition of a correlation matrix converted from an estimated covariance matrix with the lowest possible determinant computed using the minimum covariance determinant (MCD) algorithm [49,50]. The covariance matrix was computed using a subroutine (mcdcov) in MATLAB (see below). The MCD estimator is considered to be a highly robust estimator of multivariate location and scatter.

## 2.3. Principal Component Weighted Index

The principal component weighted index was defined in consistency with the SAW model as

$$PCWI = \sum_{k=1}^{q} u_k t_k = \sum_{k=1}^{q} u_k \sum_{j=1}^{m} w_{jk} x_j \qquad (2)$$

where $u_k$ stands for the weight of $k$-th PC computed as

$$u_k = \frac{\lambda_k}{\sum\limits_{k=1}^{q} \lambda_k} \qquad (3)$$

and where $\lambda_k$ is the eigenvalue of $k$-th PC and $q$ is the number of selected principal components.

## 2.4. Statistical Calculations

The original data matrixes of 67 wastewater samples were set up and processed in MS Excel. Statistical calculations were performed using the software packages QC.Expert (Trilobyte, Pardubice, Czech Republic) and XLSTAT 2018 (Addinsoft, Boston, MA, USA). The data smoothing was performed by the fast Fourier transform (FFT) algorithm in the program OriginPro 9.0.0. (Origin Corporation, Northampton, MA, USA). The data were also standardized in order for us to avoid misclassifications arising from different orders of magnitude of variables. For this purpose, the data was mean (μ) centered and scaled by standard deviations (σ) as (x − μ)/σ. The statistical calculations were performed at the $\alpha = 0.05$ significance level.

## 3. Results and Discussion

### 3.1. Principal Component Analysis

Robust PCA was performed due to non-normal distributions of the physico-chemical parameters characterizing the wastewaters composition (see Tables 1 and 2). The eigenvalue decomposition of covariance matrixes with the lowest possible determinant was computed using the MCD algorithm. Based on the PCA results, the wastewater samples were characterized by a few first PCs and relationships between original parameters were discussed.

There is no universal rule for the estimation of a number of PCs. The first five principal components for both raw and treated wastewater were selected according to the magnitudes of corresponding eigenvalues, which should be equal to or higher than 1 [51], and according to their scree plots [52]. The

eigenvalue scree and cumulative variability plots are demonstrated in Figure S1. In both cases, the selected PCs explained 88% and 83% of the total variability of raw and treated wastewater, respectively. It agrees with another traditional common rule that the cumulative proportion of variance could be explained by at least 80% [53].

### 3.2. Interpretation of Selected Principal Components

PCA often includes the interpretation of PCs which is necessary to understand the data structure. The component loadings summarized in Tables 3 and 4 can explain relationships among the original variables (parameters). In the case of the raw wastewater, the 1st principal component (PC1) was saturated mainly by ammonium, TN, BOD, COD, phosphate, and TP. All these parameters characterize organic and inorganic compounds occurring in municipal wastewater. The 2nd principal component (PC2) was affected by nitrate and nitrite resulting from nitrification processes in raw wastewater. The 3rd principal component (PC3) was affected mostly by TSS and TDS, the 4th principal component (PC4) by BOD and TDS, and 5th component (PC5) by pH.

**Table 3.** Loadings of selected principal components of raw wastewater.

| Parameters | PC1 | PC2 | PC3 | PC4 | PC5 |
|---|---|---|---|---|---|
| $NH_4^+$ | 0.912 | 0.111 | −0.166 | −0.068 | −0.007 |
| BOD | 0.678 | 0.133 | 0.158 | 0.525 | 0.287 |
| COD | 0.897 | 0.164 | 0.130 | 0.187 | 0.061 |
| $NO_3^-$ | −0.362 | 0.809 | 0.153 | −0.124 | 0.094 |
| $NO_2^-$ | −0.393 | 0.741 | 0.137 | −0.273 | 0.252 |
| $PO_4^{3-}$ | 0.867 | 0.040 | −0.042 | −0.314 | −0.296 |
| TN | 0.913 | 0.164 | −0.033 | −0.103 | 0.057 |
| TSS | 0.553 | −0.129 | 0.680 | 0.076 | 0.165 |
| TP | 0.843 | 0.143 | 0.117 | −0.304 | −0.270 |
| pH | 0.434 | −0.293 | −0.448 | −0.338 | 0.622 |
| TDS | 0.285 | 0.487 | −0.621 | 0.397 | −0.135 |

**Table 4.** Loadings of selected principal components of treated wastewater.

| Parameters | PC1 | PC2 | PC3 | PC4 | PC5 |
|---|---|---|---|---|---|
| $NH_4^+$ | −0.408 | 0.279 | 0.760 | 0.041 | 0.123 |
| BOD | −0.405 | 0.710 | 0.151 | −0.016 | −0.246 |
| COD | 0.390 | 0.561 | −0.022 | 0.018 | 0.150 |
| $NO_3^-$ | 0.812 | 0.210 | −0.281 | −0.117 | −0.369 |
| $NO_2^-$ | −0.282 | 0.410 | 0.562 | −0.329 | −0.267 |
| $PO_4^{3-}$ | 0.809 | −0.128 | 0.438 | −0.179 | 0.263 |
| TN | 0.828 | 0.273 | 0.114 | −0.020 | −0.334 |
| TSS | −0.399 | 0.568 | −0.309 | −0.358 | 0.390 |
| TP | 0.823 | −0.116 | 0.406 | −0.167 | 0.286 |
| pH | −0.335 | −0.572 | 0.532 | 0.103 | −0.188 |
| TDS | 0.265 | 0.509 | 0.145 | 0.737 | 0.141 |

In the case of the treated wastewater, PC1 was mainly affected by nitrate, phosphate, and also by TN and TP, that is, by nutrients which went through the aerobic part of an activation tank. PC2 was affected by BOD and COD, characterizing the content of organic compounds which were persistent to the treatment process. PC3 was mainly saturated by ammonium and pH, indicating the presence of un-oxidized ammonium under acidic conditions during nitrification process as follows

$$2NH_4^+ + 3O_2 \rightarrow 2NO_2^- + 4H^+ + 2H_2O \qquad (4)$$

$$2NO_2^- + O_2 \rightarrow 2NO_3^- \qquad (5)$$

PC4 and PC5 were saturated by TDS (PC4), nitrate, and TSS (PC5) which were of low concentrations and thus contributed to the less important PCs.

### 3.3. Principal Component Weighted Index

Considering the fact that scores of individual PCs have different variability depending on their eigenvalues, both PCWIs were composed of the five weighted PCs and plotted in Figure 1. Since the samples were taken approximately monthly, the vertical axis (Sample) also presents the time axis. The plots were also smoothed by the FFT procedure (red and blue curves) for us to clearly see the temporal wastewater quality changes. Approximately six-month cycles were observed and also confirmed by the time series analysis performed using the seasonal autoregressive integrated moving average model (SARIMA). The PCWI values corresponding to the raw wastewater were slowly elevated in time. During the first 48 months they were lower than those of treated wastewater and then became higher. The reason is that new buildings were connected to a local sewage system and wastewater pollution increased. The PCWI of treated wastewater continually elevated up to the 45th month and then changed, similar to the case of the raw wastewater. The similarities between both PCWI values are discussed below.

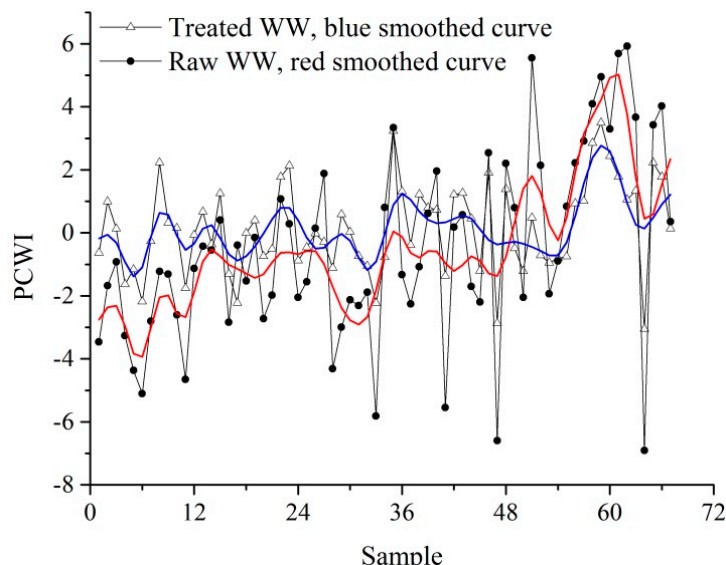

**Figure 1.** Principal component weighted index (PCWI) plots of raw and treated wastewaters.

### 3.4. Validation of PCWI

The normal distributions of the raw and treated wastewater PCWIs were confirmed by several common statistical tests, such as the Kolmogorov–Smirnov test ($p = 0.523$ and $p = 0.741$, respectively), the Shapiro–Wilk test ($p = 0.506$ and $p = 0.915$), Anderson–Darling test ($p = 0.415$ and $p = 0.863$), and Jarque–Bera test ($p = 0.730$ and $p = 0.811$). Their normality was also documented by their skewness and kurtosis of 0.153 and 0.107, resp. 2.64 and 2.64 for the raw and treated wastewater. Their standard deviations of 2.992 and 1.437 for raw and treated wastewaters, respectively, were consistent with the summary statistics given in Tables 1 and 2.

Although the normality was confirmed, the PCWIs were further analyzed by means of the Gaussian mixture modelling using the iterative EM algorithm [54]. The number of mixtures (groups) was determined according to the Bayesian information criterion, the Akaike information criterion, the Integrate complete likelihood, and the Normalized entropy criterion. Figure 2 displays the PCWI density functions corresponding to the raw (Figure 2a) and treated wastewater samples (Figure 2b) separated into two smaller groups. One group of the raw wastewater samples consisted of the samples 1–34 with the lower PCWIs and the second group consisted of the samples 35–67 with the higher

PCWIs in agreement with their temporal changes mentioned above. A similar temporal effect was found for the PCWI values of treated wastewaters. One group was mostly composed of the samples 1–45 and the second one of the samples 46–67. Such temporal behavior of the PCWIs was already demonstrated in Figure 1.

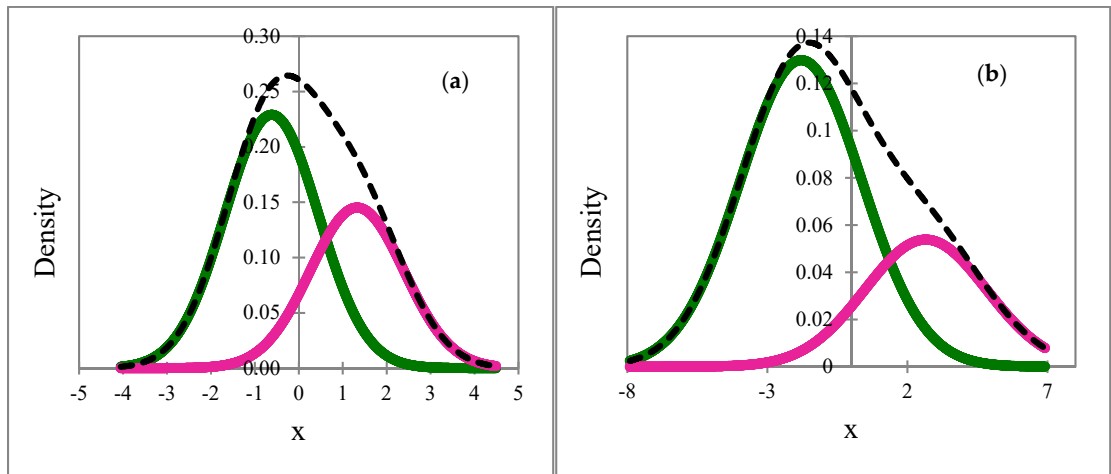

**Figure 2.** Gaussian mixture models of PCWI of raw (**a**) and treated (**b**) wastewater.

The similarity between the PCWI values of raw and treated wastewater was also confirmed by Pearson's and Spearman's correlation coefficients of 0.761 and 0.743, respectively, indicating their significant correlation demonstrated in Figure 3. The monitored BWWTP treated municipal wastewater was coming from households and non-industrial institutions which is why no unexpected pollution was supposed.

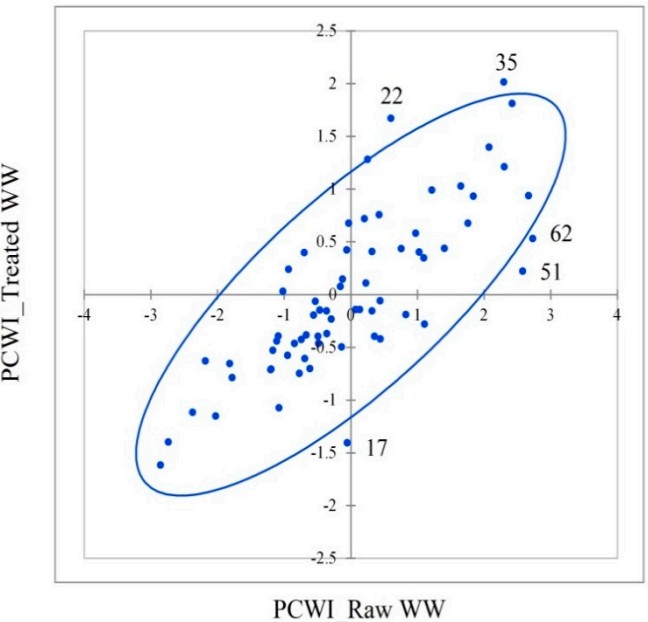

**Figure 3.** Scatter plot of PCWI scores with confidence ellipse. The labeled samples 17, 22, 35, 51, and 62 were out of confidence interval (97.5%, Chi-square).

However, five samples 17, 22, 35, 51, and 62 lying out of the confidence ellipse indicated some deviations from the steady-state treatment process. For example, the sample 35 was characterized by the high PCWIs corresponding to both raw and treated wastewater. This demonstrates a situation when the raw wastewater was treated very effectively in terms of BOD and COD but not in terms

of TN and TDS. The concentrations of nitrite and TDS were too high in the raw wastewater and the concentrations of ammonium, TN, BOD, and COD were too high in the treated wastewater, likely due to some problems in the treatment technology. The physico-chemical parameters of these outlying samples are summarized in Table S1. Individual parameters were assessed by means of the Box and Whisker plots.

There is no "gold" standard composite index which could serve for the PCWI verification. Therefore, the PCWIs were verified based on their relationships with the individual physico-chemical parameters. One example concerning COD is shown in Figure 4. COD is the common parameter used for characterization of the total content of organic and inorganic compounds which can be oxidized by potassium dichromate. Since the COD values in raw and treated wastewaters are very different, their standardized ones were plotted in one graph. The temporal changes of the standardized CODs were similar to those of the PCWI values. The six-month periods, as well as their elevation concerning the raw wastewaters, were observed. The probable reason was already mentioned in case of the PCWI.

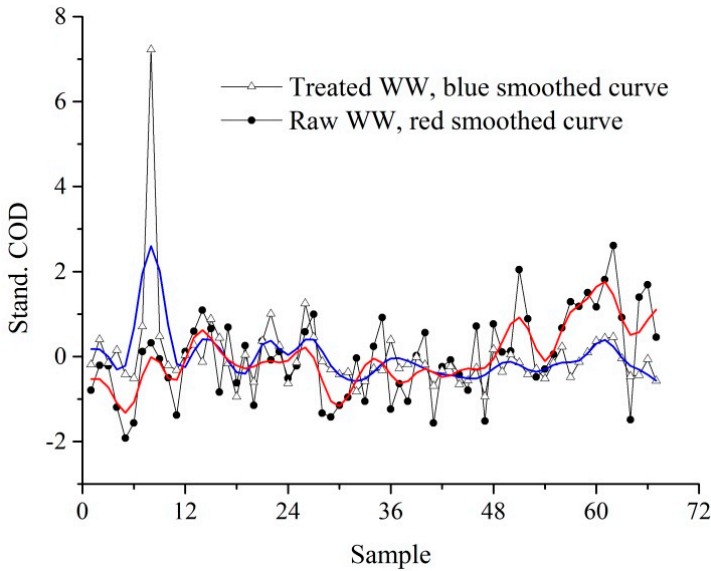

**Figure 4.** Plots of standardized COD values of raw and treated wastewater.

The similarities between the PCWIs and the original parameters were also documented by their correlation coefficients summarized in Tables S2 and S3. In the case of the raw wastewater, the PCWI significantly correlated with the parameters except nitrite, pH, and TDS. Nitrite was of low concentration, the pH changed very little, and the TDS changed independently on all the parameters except BOD. In case of the treated wastewater, the correlations were also insignificant for the parameters occurring in low concentrations, such as ammonium, BOD, nitrite, and TSS. The low concentrations of BOD corresponded to the high treatment efficiency of 95% declared by the BWWTP designer.

Comparison of PCWI with WQI

The validation of PCWI was also performed by its comparison with commonly used water quality index (WQI) which is defined as follows

$$WQI = \frac{\sum_{i=1}^{n} C_i P_i}{\sum_{i=1}^{n} P_i} \tag{6}$$

where $C_i$ and $P_i$ are the normalized values and relative weights assigned to each parameter $i$. The normalization was performed by dividing the values of each parameter by its maximal one. The relative weights ranged from 1 to 4 according to their importance for an aquatic system, which means that they are subjective: $P_i = 4$ for TSS; $P_i = 3$ for $NH_4^+$, BOD, and COD; $P_i = 2$ for TDS, TN, $NO_3^-$, and

$NO_2^-$; $P_i = 1$ for TP, $PO_4^{3-}$, and pH. They were adopted from several papers dealing with assessment of surface waters [23,27,30]. Figure 5 shows the correlation of PCWI and WQI, indicating a good agreement between both indexes. It is possible to emphasize that, unlike WQI, PCWI works with objective weights computed for particular water composition.

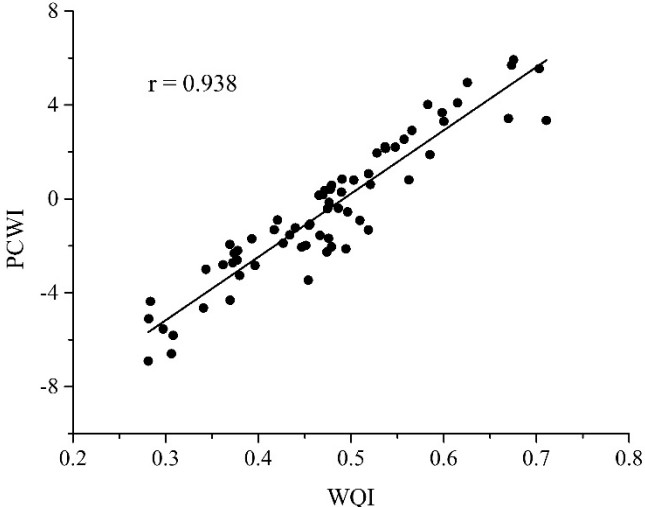

**Figure 5.** Comparison of PCWI and water quality index (WQI) computed for data of raw wastewater.

These relative weights were also used for computing WQI concerning the treated wastewater but the correlation was very weak. The weights were likely not appropriate for this type of water, but the more suitable ones were not found in the literature.

The normal distribution of WQI was also confirmed by the Kolmogorov–Smirnov test *(p = 0.607)*, Shapiro–Wilk test (*p* = 0.266), Anderson–Darling test (*p* = 0.372), and Jarque–Bera test (*p* = 0.694). The normality was also documented by a skewness of 0.212 and a kurtosis of 2.71. Sensitivity of PCWI and WQI to detect outlying observations was compared using the Grubbs test. The Z-scores of PCWI and WQI were plotted for all samples and those above and/or below +2 and −2 were detected as outliers. Figure 6 shows that using the PCWI Z-scores five samples (47, 51, 61, 62, and 64) were detected, and using the WQI ones only two samples (35 and 51) were identified. These results demonstrate that PCWI is more sensitive for the identification of anomalies in water composition.

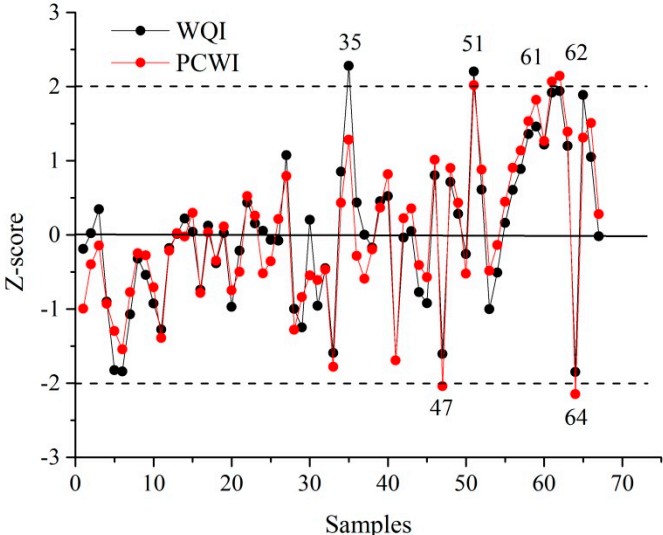

**Figure 6.** Comparison of PCWI and WQI computed for data of raw wastewater.

### 3.5. Examples of Possible PCWI Applications

As already mentioned, the PCWI has the potential to describe wastewater quality depending on time. The first example of a possible application concerns the evaluation of seasonal raw wastewater quality in various years as displayed in Figure 7.

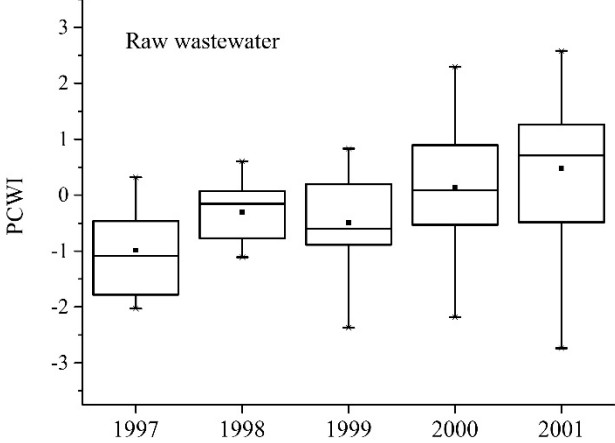

**Figure 7.** PCWI Box and Whisker plots of raw wastewater over several years. Central lines indicate medians and small rectangle symbols indicate means.

A non-parametric Kruskal–Wallis test was performed to compare the PCWIs during 1997–2001. The computed *p*-values summarized in Table 5 indicate that the wastewater quality in 1997 and 2001 (*p* = 0.031) was different. It was confirmed by the median concentrations of ammonium, TN, and COD (see Table S4).

**Table 5.** Probabilities of the Kruskal–Wallis test applied on PCWI of raw wastewater.

|      | **1997** | **1998** | **1999** | **2000** | **2001** |
|------|------|------|------|------|------|
| 1997 | 1 | 0.050 | 0.657 | 0.115 | **0.031** |
| 1998 | 0.050 | 1 | 0.888 | 0.947 | 0.347 |
| 1999 | 0.657 | 0.888 | 1 | 0.599 | 0.142 |
| 2000 | 0.115 | 0.947 | 0.599 | 1 | 0.674 |
| 2001 | **0.031** | 0.347 | 0.142 | 0.674 | 1 |

Note: Probabilities in bold are not different from 0 with a significance level $\alpha = 0.05$.

The second example concerns the evaluation of the PCWI values according to warning and control limits in analogy to the Shewhart control charts [55,56]. Upper (UWL) and lower warning limits (LWL), and upper (UCL) and lower control limits (UCL) were computed as $\mu \pm 2\sigma$ and $\mu \pm 3\sigma$, respectively (see Table 6). A majority of applications of the weighted indexes uses linear scales such as 0–25 excellent, 26–50 good, 51–75 poor, 76–100 very poor, >100 unsuitable [23,24,57]. The suggested ranks were based on the limits specific for this study.

**Table 6.** Ranking of PCWI values.

| Rank | Range | PCWI of Raw WW | N | PCWI of Treated WW | N |
|------|-------|----------------|---|--------------------|---|
| I | UCL to UWL | 8.489 to 5.495 | 3 | 4.266 to 2.890 | 2 |
| II | UWL to μ | 5.496 to −0.489 | 28 | 2.891 to 0.140 | 29 |
| III | μ to LWL | −0.490 to −6.473 | 34 | 0.141 to −2.612 | 34 |
| IV | LWI to LCL | −6.474 to −9.465 | 2 | −2.613 to −3.988 | 2 |

The number N of samples in the individual ranks are listed in Table 6. The PCWIs between UCL and UWL (rank I) as well as between LWI and LCL (rank IV) signify significant deviations from the steady-state treatment process: five samples (7.5%) of raw wastewater and four samples (6.0%) of treated wastewater. Such PCWIs ranking could be useful for operators to simply check raw and treated wastewater quality and to control working conditions on BWWTPs.

In general, the composite indexes are not supposed to be of universal validity and ability to describe reality in detail because one parameter cannot substitute a variety of variables [17,21]. Despite this, the PCWI can be employed as a useful indicator, providing overall information about water quality depending on the type of wastewater and temporal (seasonal) effects.

## 4. Conclusions

The wastewater quality before and after treatment was characterized by the principal component weighted index constructed as the sum of weighted PCs scores. The robust PCA of the 67 raw and treated wastewater samples extracted five principal PCs explaining 88%, resp. 83% of the total data variability. Based on the PCs loadings, the relationships among the original parameters were discussed. The PCWIs plots were constructed to show the temporal water quality changes. The six-month PCWIs cycles were identified. Using the Gaussian mixture modelling the PCWI values were separated into two groups of samples in agreement with the PCWI temporal plots. The PCWIs scatter plot identified the samples that deviated from the steady-state treatment. PCWI and WQI computed for the raw wastewater were compared and found to be in good agreement.

The possible application of PCWI for the raw wastewater quality monitoring was demonstrated by the evaluation of wastewater quality during 1997–2001 using a non-parametric Kruskal–Wallis test. The years 1997 and 2001 were found to be different which was explained comparing the median concentration of ammonium, BOD, and COD. The PCWI application in analogy with the Shewhart warning and control limits was also demonstrated. The PCWI was found to be used for the overall characterization of wastewater quality, especially from the temporal point of view.

**Supplementary Materials:** The following are available online at http://www.mdpi.com/2073-4441/11/11/2376/s1, Figure S1: Eigenvalue scree and cumulative variability plots for raw and treated wastewaters, Table S1: Composition of outlaying samples, Table S2: Spearman's correlation matrix of original parameters and PCWI of raw wastewater, Table S3: Spearman's correlation matrix of original parameters and PCWI of treated wastewater, Table S4: Medians of raw wastewater parameters in 1997–2001.

**Funding:** This work was financially supported by the projects "Institute of Environmental Technology—Excellent Research" (CZ.02.1.01/0.0/0.0/16_019/0000853) and Large Research Infrastructure ENREGAT (project No. LM 2018098) provided by the Ministry of Education, Youth and Sports of the Czech Republic. The author has no conflict of interest to declare.

**Conflicts of Interest:** The author declares no conflict of interest. The funders had no role in the design of the study; in the collection, analyses, or interpretation of data; in the writing of the manuscript, or in the decision to publish the results.

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
