# Peer review of "Principal Component Weighted Index for Wastewater Quality Monitoring"

_water, doi:10.3390/w11112376_

Round 1
Reviewer 1 Report
I revised the paper titled “Principal component weighted index for wastewater quality monitoring”. I think that the matter of the paper is in the scope of the Journal and I found it very interesting. The structure of the paper is good and the English is correct. It can be published on “Water” if the following minor revisions will be implemented.
My suggestions are the following:
1) Introduction: I think that this section needs to be implemented and revised. For instance, not only the mathematical aspects should be described but also the worsening of wastewater quality with associated with the release of high quantity of contaminants such as dyes, surfactants, pesticides, heavy metals… I suggest you this recent papers: (https://doi.org/10.1016/j.jenvman.2018.11.094) (https://doi.org/10.1016/j.psep.2019.10.022) (https://doi.org/10.1007/s11270-019-4158-1)
In this way, I think that the importance of calculating indices in order to monitor the quality of wastewater could be more clearly highlighted.
Moreover; in Line 39: Define PCA and in Line 43: Define PCWI.
2) Materials and methods. I think that some other information regards the WWTP could be useful. What types of treatments are adopted?
3) Figure 3. I think it is unclear the meaning of the numbers in the graph outside the ellipse. Please define in the caption.
4) Figure 4. Please, define red and blue curve in the figure and not in the caption.
5) Figure 5. What the line means? What the box means? Maybe the mean and the median but please define.
6) Results and discussion. I think that a new section (e.g. 3.6) with a comparison with recent literature results should be added. For instance, the authors could compare their results (PCWI) with other indexes calculated on the basis on the characteristics of the same wastewater in order to show the different (or similar) results.
Author Response
The author thanks very much the reviewers for their careful reading and valuable comments on his manuscript. The manuscript was revised and modified accordingly. The manuscript was completed by new text, which was highlighted in yellow colour, and two new figures. The reviewers’ comments are answered below.
Reviewer #1
I revised the paper titled “Principal component weighted index for wastewater quality monitoring”. I think that the matter of the paper is in the scope of the Journal and I found it very interesting. The structure of the paper is good and the English is correct. It can be published on “Water” if the following minor revisions will be implemented.
My suggestions are the following:
Comment #1
Introduction: I think that this section needs to be implemented and revised. For instance, not only the mathematical aspects should be described but also the worsening of wastewater quality with associated with the release of high quantity of contaminants such as dyes, surfactants, pesticides, heavy metals… I suggest you this recent papers: (https://doi.org/10.1016/j.jenvman.2018.11.094) (https://doi.org/10.1016/j.psep.2019.10.022) (https://doi.org/10.1007/s11270-019-4158-1)
In this way, I think that the importance of calculating indices in order to monitor the quality of wastewater could be more clearly highlighted.
Moreover; in Line 39: Define PCA and in Line 43: Define PCWI.
Response #1
The suggested papers as well as other papers dealing with Water quality index were included in Introduction. The PCA and PCWI were defined at their first appearance in Abstract.
Comment #2
Materials and methods. I think that some other information regards the WWTP could be useful. What types of treatments are adopted?
Response #2
A brief description of BWWTP was placed into the part 2.1
Comment #3
I think it is unclear the meaning of the numbers in the graph outside the ellipse. Please define in the caption.
Response #3
The samples 17, 22, 35, 51 and 62 were mentioned in the text and in the figure caption.
Comment #4
Please, define red and blue curve in the figure and not in the caption.
Response #4
The red and blue curves were defined in Figure 1. The description was deleted from the caption. The same change was made in Figure 4.
Comment #5
What the line means? What the box means? Maybe the mean and the median but please define.
Response #5
The inner horizontal line is a median and the small box is a mean. It was defined in the figure caption.
Comment #6
Results and discussion. I think that a new section (e.g. 3.6) with a comparison with recent literature results should be added. For instance, the authors could compare their results (PCWI) with other indexes calculated on the basis on the characteristics of the same wastewater in order to show the different (or similar) results.
Response #6
A new sub-section 3.4.1 was written. The comparison of PCWI and standard WQI was performed by linear regression shown in new Figure 5.
Reviewer 2
This seems a reasonable model approach to study the development of a simplified index. I am not so much of an expert in statistical assessment of WWT processes and effect.
Comment #1
However it seems there may be issues where there are challenges to the process and spikes in parameters from the unexplained change in input? Comparison of in an out values over time v new index. Sensitivity of index?
Response #1
The new section 3.4.1 was added into the text. A part of this section was devoted to the sensitivity of the indexes for the detection of outliers. Using Z-score plots, PCWI was demonstrated to be more sensitive than WQI.

Reviewer 2 Report
this seems a reasonable model approach to study the development of a simplified index. i am not so much of an expert in statistical assessment of WWT processes and effect. however seems there may be issues where there are challenges to the process and spikes in parameters from the unexplained change in input? comparison of in a out values over time v new index. sensitivity of index?
Author Response

(The authors gave the same response as above.)
